# Genetic and Epigenetic Signatures Associated with the Divergence of *Aquilegia* Species

**DOI:** 10.3390/genes13050793

**Published:** 2022-04-28

**Authors:** Zhenhui Wang, Tianyuan Lu, Mingrui Li, Ning Ding, Lizhen Lan, Xiang Gao, Aisheng Xiong, Jian Zhang, Linfeng Li

**Affiliations:** 1Faculty of Agronomy, Jilin Agricultural University, Changchun 130118, China; wzhjlau@163.com; 2Ministry of Education Key Laboratory for Biodiversity Science and Ecological Engineering, School of Life Sciences, Fudan University, Shanghai 200438, China; tianyuan.lu@mail.mcgill.ca (T.L.); limingrui@fudan.edu.cn (M.L.); 14307110009@fudan.edu.cn (N.D.); 13484293951@163.com (L.L.); 3Genome Quebec Innovation Center, McGill University, Montreal, QC H3A 0G1, Canada; 4Lady Davis Institute, SMBD JGH, Montreal, QC H3A 1A3, Canada; 5Key Laboratory of Molecular Epigenetics of the Ministry of Education, Northeast Normal University, Changchun 130024, China; gaoxiao424@163.com; 6State Key Laboratory of Crop Genetics and Germplasm Enhancement, Ministry of Agriculture and Rural Affairs, Key Laboratory of Biology and Germplasm Enhancement of Horticultural Crops in East China, College of Horticulture, Nanjing Agricultural University, 1 Weigang, Nanjing 210095, China; xiongaisheng@njau.edu.cn; 7Department of Biology, University of British Columbia, Okanagan, Kelowna, BC V1V 1V7, Canada

**Keywords:** adaptive radiation, *Aquilegia*, selection, whole-genome sequencing, DNA methylation

## Abstract

Widely grown in the Northern Hemisphere, the genus *Aquilegia* (columbine) is a model system in adaptive radiation research. While morphological variations between species have been associated with environmental factors, such as pollinators, how genetic and epigenetic factors are involved in the rapid divergence in this genus remains under investigated. In this study, we surveyed the genomes and DNA methylomes of ten *Aquilegia* species, representative of the Asian, European and North American lineages. Our analyses of the phylogeny and population structure revealed high genetic and DNA methylomic divergence across these three lineages. By multi-level genome-wide scanning, we identified candidate genes exhibiting lineage-specific genetic or epigenetic variation patterns that were signatures of inter-specific divergence. We demonstrated that these species-specific genetic variations and epigenetic variabilities are partially independent and are both functionally related to various biological processes vital to adaptation, including stress tolerance, cell reproduction and DNA repair. Our study provides an exploratory overview of how genetic and epigenetic signatures are associated with the diversification of the *Aquilegia* species.

## 1. Introduction

Adaptive radiation is the rapid diversification of a single ancestral species into a vast array of common descendants that inhabit different ecological niches or use a variety of resources but differ in phenotypic traits required to exploit diverse environments [1,2,3,4]. Disentangling the evolutionary mechanisms that underpin adaptive radiation is fundamental to understanding the evolution and persistence of biodiversity [5,6]. This has been a key focus of many studies investigating different animal and plant lineages that diversified through adaptive radiation, including Hawaiian silvers word, Caribbean anoles, Darwin’s finches, and African cichlids [7,8,9,10]. In past decades, accumulating evidence from diverse lineages that have radiated suggests that both extrinsic environmental factors (e.g., resource availability) and genetic variation can determine the rate and volume of species diversification [11]. Among the environmental factors involved, ecological opportunity is considered as the primary mechanism that causes rapid adaptive radiation through the acquisition of key innovations, invasion of new environments, and extinction of competitors [2,12]. On the other hand, new species also arise as a result of new genetic variation being preserved among closely related species that ultimately influences phenotypic disparity, upon which natural selection acts [13]. In the rapid speciation of African cichlid fishes, extrinsic environmental factors (e.g., ecological specialization) and genetic mechanisms (e.g., adaptive introgression) acted together to provoke the repeated adaptive radiation in geographically isolated lakes [10,11,14,15]. On the other hand, the evolutionary roles of epigenetic modification in plant adaptation have also been discussed for decades [16,17,18]. For example, genome-wide comparisons of the methylation pattern among the Arabidopsis accessions, which were inherited from the same mother individual over 30 generations, clearly showed that differential cytosine methylation at specific sites is associated with phenotypic diversity [19,20].

The genus *Aquilegia* L. (columbine) is a well-recognized model system for the study of the evolutionary mechanisms underlying adaptive radiation [21,22]. This genus includes approximately 70 recently diversified species that are widely distributed in the temperate zones of North America and Eurasia [23]. Phylogenetic and geographic inferences have illustrated two independent adaptive radiations of North American and European lineages from the ancestral Asian species [21,24]. For example, floral diversification of the North American *Aquilegia* species is highly correlated with the pollinator specialization [25,26,27,28]. In contrast, ecological adaptation and geographic isolation are considered as the major driving forces that promoted rapid radiation of the European species [21,29]. In Asia, different pollinators and ecological habitats are both proposed to have resulted in the diversification of more than 20 morphologically distinct species [30,31]. These Asian *Aquilegia* species constitute four highly divergent lineages corresponding to their geographic origins, and have evolved relatively independently [30,31]. Despite this well-described evolutionary history and the crucial role played by environmental factors, how genetic and epigenetic factors are involved in the rapid speciation in this genus remains poorly investigated.

In this study, we surveyed the genomes and DNA methylomes of 36 accessions from 10 worldwide *Aquilegia* species and have systematically revealed their species-specific genetic and epigenetic features. We have also comprehensively identified and functionally characterized the genes harboring adaptation-associated genetic and epigenetic variations, providing a genome-wide view of the molecular variation patterns associated with the diversification of *Aquilegia* species.

## 2. Materials and Methods

### 2.1. Sample Collection, DNA Extraction and Whole-Genome Sequencing

In this study, a total of 36 accessions from 10 worldwide *Aquilegia* species were collected (Appendix A). Among the Asian species, four phylogenetically distinct species (*A. japonica*, *A. oxysepala, A*. *yabeana*, and *A*. *viridiflora*) were selected according to their geographic distributions and ecological habitats. *A. japonica* and *A. oxysepala* are sister species that inhabit alpine tundra and low-altitude forest niches in northeastern China, respectively [31,32]. Eighteen accessions were collected to represent these two Asian species and their putative hybrids. In addition, four accessions were collected from the other two Asian species, *A*. *yabeana* and *A*. *viridiflora*. The former species shares highly similar morphological traits and ecological niches with *A. oxysepala,* but is allopatrically distributed in northern China. In contrast, *A. viridiflora* is sympatrically distributed with *A*. *yabeana* and *A. oxysepala* in northern and northeastern China, but often occupies rocky and sandy ecological niches. Furthermore, six and eight accessions were sampled from the European and North American lineages, respectively. All the 36 accessions were grown in a greenhouse under the same conditions (25 °C/12 h, 16 °C/12 h). Mature leaves of each species were collected from each of these accessions at the same developmental stage. Genomic DNA was extracted from fresh mature leaves using a TianGen plant genomic DNA kit. Whole-genome resequencing and bisulfite sequencing were performed on the extracted genomic DNA using the Illumina X-ten platform (Illumina, CA, USA). Short-insert (350 bp) DNA libraries of all the accessions were constructed by NovoGene (NovoGene, Tianjin, China). The genome assembly of an admixed species, *A. coerulea* “Goldsmith”, was obtained from Phytozome v12.1 (https://phytozome.jgi.doe.gov, accessed on 25 January 2019) as the reference genome [22].

### 2.2. Sequence Assembly, Functional Annotation and Genetic Diversity

Whole-genome sequences of each accession were aligned against the reference genome, using the default settings of the BWA-MEM algorithm implemented in the Burrows–Wheeler Aligner (BWA) [33]. Raw assemblies were realigned using IndelRealigner provided in the Genome Analysis Toolkit by default settings [34]. Single nucleotide polymorphisms (SNPs) and insertions/deletions (INDELs) were reported using SAMtools [35]. Only the high-quality variants (SNPs and INDELs) (read depth of >3, mapping quality of >20 and missing allele of <1%) were retained for subsequent population genomics analyses. Genomic annotation of the identified variants was reported for each of the 36 samples separately. Functional annotation of each identified variant was performed using SnpEff, based on the reference genome [36].

To infer the phylogenetic relationship between the ten *Aquilegia* species, NJ trees were reconstructed for each chromosome and the whole-genome dataset, using MEGA 7 [37]. A principal component analysis (PCA) was carried out to examine the genetic diversity of the 36 *Aquilegia* accessions [38]. The ancestral components were estimated using ADMIXTURE [39], with different numbers of populations ranging from one to ten. The optimal population composition with the lowest 5-fold cross-validation error was selected to decompose the ancestral admixture. To obtain the genome-wide nucleotide variation pattern, nucleotide diversity (π) and genetic differentiation (Weir and Cockerham’s F_ST_) were calculated for each 100 kb non-overlapping sliding window, using VCFtools [40,41]. Pair-wise non-synonymous-to-synonymous (d_N_/d_S_) ratios of the ten species were inferred by the yn00 program in the Phylogenetic Analysis by Maximum Likelihood (PAML) package [42]. An inter-lineage d_N_/d_S_ value for each gene was derived by averaging the d_N_/d_S_ values obtained from all the pair-wise comparisons of the samples belonging to the two lineages under investigation. The candidate genes with the 5% highest and 5% lowest d_N_/d_S_ values were considered to have undergone strong positive and purifying selection, respectively.

### 2.3. Cytosine Methylation Pattern and Epigenetic Population Structure

Whole-genome bisulfite sequencing data were pre-processed using TrimGalore (https://www.bioinformatics.babraham.ac.uk/projects/trim_galore/, accessed on 21 August 2018). The paired-end reads were then aligned to the reference genome using Bismark [43] with a moderately stringent minimum-score function (L, 0, −0.3). De-duplicated alignments of the 36 *Aquilegia* accessions were used to report the cytosine methylation levels using bismark_methylation_extractor on loci with a read depth of ≥3. Genomic annotations of the methylated cytosine site were identified based on the reference genome, using an in-house Python script. PCA was conducted for 588,659 loci that passed the quality control to infer the CG-methylomic diversity of the ten *Aquilegia* species. Differential cytosine methylation was determined at the gene and chromosome levels, respectively. At the gene level, we determined DMRs for each 100 bp non-overlapping sliding window using the Cochran–Mantel–Haenszel (CMH) test to account for imbalanced read depth (Appendix A). The genomic regions that possessed a Benjamini–Hochberg adjusted *p* value < 0.05 and showed inter-specific or inter-lineage methylation divergence higher than 25% were defined as significant DMRs. The genes with >20% of the genic region being differentially methylated regions (DMRs) were defined as differentially methylated genes (DMGs). The chromosome level methylation patterns were measured using the chromosomal methylation discrepancy index (MDI) [44]. The methylation patterns of the identified DMGs were visually verified on Integrative Genomics Viewer [45], prior to the downstream analyses and biological interpretation. In addition, we identified CG islands from the *A. coerulea* “Goldsmith” reference genome, using EMBOSS cpgplot with default settings [46]. Only the identified CG-enriched genomic regions with > 200 bp were defined as CG islands. We then investigated the inter-specific and inter-lineage methylation patterns in and around the CG islands.

### 2.4. Associations between the Genetic Variation and Cytosine Methylation

We tested for associations between the identified DMGs and genes under positive selection by a Chi-square test. A linear regression model was adopted to measure the direct causal effect of CG-loss variation on CG methylation. To further assess whether genetic variations drive the establishment of DMG, driving mutations of DMRs between the *A. japonica* and *A. oxysepala* were identified using an Eigenstrat-based method (see Appendix A for more details) [47].

### 2.5. Identification of Conservative Clade-Specific Variant

Clade-specific variants (CCVs) were defined as variants that had a SnpEff-predicted “high” functional impact and that were conserved across all the samples belonging to the same species or lineage, but not present in any sample of the other species/lineages. Since the biological consequences of heterozygous variants were less affirmable, only the homozygous point mutations and INDELs were included in the characterization of CCVs, including frameshift, stop-gain, stop-loss, start-loss and splicing-alteration variations. 

### 2.6. Functional Analysis

The aforementioned genetic and epigenetic analyses identified candidate genes that might be associated with the rapid diversification of the *Aquilegia* species from different perspectives. These candidate genes were employed to conduct functional enrichment analyses using the R package topGO with default settings [48]. The enriched GO terms that possessed a *p* value of <0.05 were considered to be statistically significant. Since the statistical tests performed by topGO are not independent, multiple testing correction does not apply [48]. The structures of the functional domains of targeted genes were determined based on the InterPro database (https://www.ebi.ac.uk/interpro, accessed on 25 January 2019). The distribution patterns of the identified candidate genes and their related functional pathways were visualized using the R package jvenn [49].

## 3. Results

### 3.1. Population Structure and Nucleotide Variation Pattern

Neighbor-joining (NJ) trees were reconstructed for 36 accessions, representing 10 *Aquilegia* species based on 689,123 homozygous SNPs. The phylogeny showed that these accessions formed three distinct lineages corresponding to their geographic origins (Figure 1a and Appendix A). In brief, all twenty-two accessions of the four East Asian species, *A. japonica*, *A. oxysepala*, *A*. *yabeana* and *A*. *viridiflora*, clustered as a monophyletic lineage, with the first two species and their hybrids forming a clade and the last two species grouping as a sister clade. In contrast, the West Asian species *A. fragrans* clustered with the geographically adjoining European species. Within the North American lineage, phylogenetic relationships of the four species varied across the seven chromosomes. In particular, one European accession of the *A. alpina* var. *alba* had a closer genetic distance to the North American lineage on chromosomes 1 and 3. Likewise, a principal component analysis (PCA) and population structure inference also revealed the distinct genetic structure of the three phylogenetic lineages (Figure 1b,c). Consistent with the above phylogeny, one *A. alpina* var. *alba* accession shared the same ancestral genetic cluster with the North American lineage, while the putative hybrids of the *A. oxysepala* and *A. japonica* possessed an admixed genetic background (Figure 1b,c).

To gain further insight into the genome-wide nucleotide variation pattern of the ten *Aquilegia* species, we calculated the nucleotide diversity (π) and genetic divergence (F_ST_) for each chromosome and for the 100-kb sliding windows. Among the three phylogenetic lineages, the Asian *Aquilegia* species harbored the highest nucleotide diversity, compared to the European and North American lineages across the seven chromosomes (Figure 2). By comparing the nucleotide diversity for each 100-kb sliding window, we observed a moderate correlation of a genome-wide variation pattern among the three lineages (Spearman R = 0.42–0.56) and a high correlation between *A. oxysepala* and *A. japonica* (Spearman R = 0.70) (Appendix A). In particular, 116 of 241 low genetic diversity genomic regions (LDGRs, with 5% lowest π) were shared by at least two of the three lineages (Appendix A). Between *A. oxysepala* and *A. japonica*, while we defined 148 LDGRs and 148 high divergence genomic regions (HDGRs, with 5% highest F_ST_), only 7 candidate genomic regions overlapped (Appendix A). 

### 3.2. Identification of the Genomic Regions Indicating Selection Pressure and Highly Impactful Genetic Variations

The candidate genes or genomic regions associated with adaptive divergence were determined from three perspectives. First, we considered the genes localized within the regions that showed low intra-specific diversity, but high inter-specific divergence, to be representative of intra-specific genetic differences. We, thus, identified twenty-three genes from the above seven candidate genomic regions that were both HDGRs and LDGRs shared by *A. oxysepala* and *A. japonica* (Appendix A). The genes within these genomic regions were functionally associated with meiotic nuclear division, adenine methyltransferase activity and basic cellar activities.

While the first strategy mainly relied on genome-wide scanning using a 100-kb non-overlapping sliding window, we also employed a functional annotation-based approach to identify the highly impactful conservative clade-specific variations (CCVs) from both within- and between-lineage comparisons. Our results revealed that a considerable proportion (17.9–40.5%) of the CCVs were identified in gene body regions (Appendix A). We then examined the potential functional impacts of the genes harboring these identified CCVs. Between the *A. oxysepala* and *A. japonica*, the CCV-carrying genes were enriched in several vital biological pathways related to cell reproduction, including telomere maintenance, DNA repair, and DNA helicase activity (Figure 3 and Table 1). For example, two candidate genes (*Aqcoe6G160300* and *Aqcoe7G062500*) that code for *Xklp2* (*TPX2-* Targeting protein for Xklp2) were functionally correlated with spindle assembly during the mitotic process [50,51]. Among the three phylogenetic lineages, the CCV-harboring genes were also functionally involved in the mitotic chromosome condensation, DNA ligase activity and aminopeptidase activity (Figure 3 and Table 1). For instance, two CCV-containing genes (*Aqcoe2G276600* and *Aqcoe1G273400*) that encode the DNA mismatch repair proteins *MutS*/*MSH* (MutS homolog) and *MutS2* [52] carried one Asian-specific-to-American frameshift variant.

Thirdly, we also derived pair-wise synonymous (d_S_) and non-synonymous (d_N_) mutation rates to identify the genes under positive or purifying selection pressure. We found that the species within the Asian lineage experienced significantly stronger positive (d_N_/d_S_ > 1) and purifying (d_N_/d_S_ < 1) selection pressure, compared to the European and North American lineages (Wilcoxon rank sum test, all Bonferroni-corrected *p* values < 1.5 × 10^−16^) (Appendix A). Likewise, the European species showed significantly stronger purifying selection (Wilcoxon rank sum test, Bonferroni-corrected *p* value = 7.8 × 10^−8^), compared to the North American species.

### 3.3. Cytosine Methylation Patterns and Differentially Methylated Genes

In parallel with the above genomic analyses, we also investigated the CG methylation patterns of the representative *Aquilegia* species. Despite variability across the 36 *Aquilegia* accessions, the North American, Asian and European species showed no distinguishable differences (*t* test, all Bonferroni-corrected *p* values > 0.01) in the overall percentage of methylated cytosines (Figure 4a). We then performed a PCA to examine the CG-cytosine methylomic diversity of all the *Aquilegia* accessions. The resulting overall methylation pattern highly resembled the SNP analysis, with the European and American species forming two distinct groups and the four Asian species forming three separate clusters (Appendix A). We then assessed the CG methylation patterns for the European and North American lineages, as well as the three Asian species (*A. japonica*, *A. oxysepala* and *A. viridiflora*) separately. Consistent with the described genomic features, a heterogeneous CG methylation pattern was also observed for the seven chromosomes, with chromosome 4 demonstrating an obviously higher overall CG methylation divergence, compared to the other chromosomes (Figure 4b). We further quantified the CG methylation level in the genic regions, putative cis-regulatory regions and CG islands. In the genic and regulatory regions, all three lineages shared similar modification patterns, with an apparent depletion of CG methylation around the transcription start site (TSS) and transcription end site (TES) (Figure 4c). However, the American lineage exhibited hyper-methylation (more than 10%) around the center of the CG islands and a more drastic decrease throughout the CG island shores, compared to the European and Asian species (Figure 4d).

To examine the biological impacts of CG methylation on species diversification, differentially methylated regions (DMRs) and differentially methylated genes (DMGs) were identified for both within- and between-lineage comparisons (Appendix A). Within the Asian lineage, 3622 DMRs in 2899 DMGs were identified between *A. japonica* and *A. oxysepala*. The functional enrichment of these DMGs indicated that the two species may have different activities in the photosynthesis-related pathways, including photosystem I, photosynthesis and chloroplast (Figure 3). For example, two photosynthesis-related genes, PsaA/PsaB and CemA, showed significantly differential methylation between the two species in the genic regions (Appendix A). At the inter-lineage level, more DMGs were identified between the North American and European species (6087 genes), compared to those between the two lineages and the Asian species (3308–5003 genes) (Appendix A). The DMGs characterized from the inter-lineage comparisons were mainly involved in plant growth (e.g., response to auxin) and defense (e.g., response to biotic stimulus and wounding) (Figure 3).

We then examined whether the candidate genes (CCV-carrying genes and DMGs) superimposed on the same signature of natural selection. We found that, while a considerable proportion of the candidate genes were shared for each of the genetic- and epigenetic-based assessments (Appendix A), they showed a segregated distribution pattern across all the comparisons (Appendix A). Likewise, the Gene Ontology (GO) enrichment analyses of the candidate genes identified from the genetic and epigenetic analyses were enriched in functionally complementary pathways (Figure 3), suggesting the co-existence of different underlying evolutionary mechanisms.

### 3.4. Association between Epigenetic Genetic and Variation

Since both genetic variation and differential CG methylation seemed to have crucial and multifaceted influences on the adaptation of the ten *Aquilegia* species, we wondered whether differential epigenetic modifications were dependent on genetic variation. Among the 588,659 CG loci examined, 224,222 (38.09%) carried a CG-loss variation. We then examined the epigenetic variation for the variant and non-variant CG loci. As shown in Figure 5, genetic-epigenetic associations of varying magnitude were observed in both types of CG loci. The variation-carrying CG loci conveyed information that highly resembled their genetic background. The overall methylation pattern was highly conserved within the same species, but exhibited obvious divergence across the ten *Aquilegia* species (Figure 5a). In contrast, CG methylation divergence at the non-variant CG loci showed higher variability at both the intra- and inter-specific levels (Figure 5b). By examining the correlation of the genetic variability and cytosine methylation, we found that CG methylation divergence at the variation-carrying CG site was largely attributable to the CG-loss variations (Figure 5c). In particular, 75% of the CG-loss variations that occurred at the most highly variable CG-methylated dinucleotides could explain at least 75% of the total epigenetic variability *per se*. Nevertheless, there was still a considerable proportion of epigenetic variability that could not be sufficiently explained by the variant-CG loci (Figure 5d). 

We also attempted to identify the *cis*-driver mutations for each of the 1229 DMRs between the *A. japonica* and *A. oxysepala*. Our results revealed that only 568 out of the 1229 (46.2%) DMRs were significantly associated with at least one genetic variation inside or around a 500 base-pair (bp) upstream/downstream genomic region, even under the least stringent *p* value threshold (5 × 10^−5^), indicating that the epigenetic changes were only partially dependent on *cis*-genetic driving mutations (Figure 5e). Moreover, we observed weak, yet significant, associations between differential CG methylation and selection pressure. In most inter-lineage comparisons, the DMGs were significantly more prone to be under positive selection than non-DMGs (Table 2), suggesting that epigenetic modifications are associated with the shaping of genotypes by selection pressure. In contrast, the DMGs were significantly less prone to be under purifying selection (Table 2).

## 4. Discussion

### 4.1. Genetically Determined Mechanisms Associated with the Rapid Diversification of Aquilegia Species

Elucidating the evolutionary mechanisms underpinning species diversification is crucial to understanding the evolution and persistence of biodiversity [2,5,6]. The genus *Aquilegia* provides an ideal system to address how diverse evolutionary mechanisms promote rapid adaptive radiation [21,22]. Although various environmental conditions related to ecological opportunities, such as shifts in pollinator and habitat, have been proposed to facilitate the evolution of reproductive isolation [27,31], the genetic basis of the rapid diversification of the *Aquilegia* species has not been explored at the genome-wide level. In this study, we surveyed the genomes of ten *Aquilegia* species to address whether specific changes in genetic architecture have been involved in the rapid species diversification. Broadly consistent with the previously inferred phylogenies [21,24,31,53], the ten *Aquilegia* species from Asia, Europe and North America formed three phylogenetically independent lineages corresponding to their geographic origins. This attribute renders the *Aquilegia* species a suitable system to identify genomic variations associated with the repeated adaptive speciation.

It has been proposed that if genetic factors promote adaptive speciation, one would expect to identify specific genetic architectures in the diversified lineages [9,14,54]. In Darwin’s finches, for example, polyphyletic topology was observed as a general pattern in 14 morphologically distinct species and phenotypic diversity of the beak shape was mainly determined by natural selection acting on the *ALX1* (aristaless-related homeobox transcription factors 1) gene during the ecological specialization process [9]. A similar phenomenon was observed in the East African cichlid fish, where the radiating lineages were more dynamic in terms of gene content and transcriptomic landscape, compared to their non-radiating relatives [14,54]. In this study, the genome-wide nucleotide variation pattern strongly reflects the allopatric evolution of these species in their respective geographic regions. This also suggests, as found for cichlid fish and the Galapagos finches, that genetic variation is intertwined with changes in the environment during the diversification process. As expected, our genome-wide scanning for selection signatures revealed distinct positive and purifying selection modes in the intra- and inter-lineage comparisons. More importantly, the CCV-carrying genes identified from the three lineages are associated with cell reproduction (e.g., telomere maintenance and mitotic chromosome condensation) and other functionally important traits.

Compared to the genetic signatures identified among the three lineages, we also observed species-specific genetic variations between the two ecological species *A. japonica* and *A. oxysepala*. Our previous studies have demonstrated that natural selection and genetic drift together resulted in the rapid divergence [31,53]. In this paper, we further demonstrate that the candidate genes involved in adaptive speciation are functionally enriched in the pathways related to cell reproduction (e.g., telomere maintenance), stress tolerance (e.g., response to wounding) and basic cellular activities. It should be noted that, although a majority of the enriched pathways are specific to each comparison, enrichment of the cell reproduction-related pathways (e.g., telomere maintenance, DNA repair and DNA helicase activity) and stress tolerance are shared in the intra- and inter-lineage comparisons. Taken together, these findings indicate that specific genetic determinants related to chromosomal architecture might have played a role in the speciation of the *Aquilegia* species, possibly underscoring the adaptations that occurred to cope with the changing environmental conditions. 

### 4.2. Associations between Cytosine Methylation and Diversification of Aquilegia Species

The role of epigenetic modification in the long-term evolutionary process has long been debated [55,56,57]. It has been proposed that epigenetic variations are frequently under genetic control, which can alter rapidly as a result of environmental induction and stochastic epimutation [58,59]. Nevertheless, it has also been recognized that some epigenetic variations can persist over generations and be highly correlated with phenotypic diversity [57]. As illustrated in Arabidopsis, changes in cytosine methylation can produce meiotically stable epialleles, which could eventually lead to phenotypic diversity in the absence of genetic variations [19,20,60]. In this paper, we assessed whether the epigenetic modifications were also associated with the adaptive speciation of the *Aquilegia* species. Consistent with the genomic features detailed above, high divergence of cytosine methylation was observed across the Asian, European and North American lineages. Notably, differential cytosine methylation was not only found across the seven chromosomes, but was also evident in the gene body of DMGs and CG island regions among the three lineages. Particularly, the functional enrichment analyses identified significant associations with adaptation-related traits, including plant growth, stress tolerance and basic cellular activities. For example, the candidate DMGs, identified between the *A. japonica* and *A. oxysepala*, showed significant enrichment in the pathways related to diverse important phenotypic traits, such as photosynthesis, embryo development and response to auxin. These features indicated that epigenetic factors might also play a role in the responses to diverse environmental conditions.

We noted that some candidate genes and enriched pathways shared hotspots of both genetic and epigenetic disparities, especially those related to cell reproduction, plant growth and stress tolerance. Many studies based on humans and mice have shown that genetic variations can affect *cis*-CG methylation at specific loci to further influence phenotypes, where CG methylation serves as a mediator [61,62]. By analyzing the associations between genetic and epigenetic variability, we conclude that, while many CG-loss variations can directly lead to the depletion of CG methylation, many DMRs are not affected by any *cis*-variations. Since gene body CG methylation in plants generally stabilizes gene expression and is positively correlated with gene expression [63,64,65,66], differential methylation in our study ostensibly indicates differing amounts of gene products. Indeed, increasing evidence from diverse species suggests that DNA sequence variation alone is not responsible for all the standing phenotypic variation in plants. A typical example of this phenomenon was observed in *Linnaria vulgaris,* which clearly showed that the switch from bilateral to radial floral symmetry is determined by the differential methylation at the Lcyc gene [67]. Likewise, the colorless non-ripening natural strain of tomato is caused by an epimutation of the SBP-box transcription factor [68]. 

Based on these attributes, together with the plausible associations between differential methylation (e.g., DMGs) and positive selection (e.g., d_N_/d_S_), we propose that epigenetic modification may be a complementary mechanism that facilitates phenotypic diversity of the *Aquilegia* species. 

### 4.3. Limitations and Future Directions

Our study has important limitations. First of all, the small sample size in our study may introduce bias and inflation of false positives, and we postulate that our findings should be interpreted carefully and considered as exploratory. When the association between genetic divergence and evolutionary events is investigated, it is impossible to deny the roles of other evolutionary forces. We acknowledge that the lineage-specific allele frequencies are possibly a consequence of genetic drift and that genetic hitchhiking may lead to the identification of candidate genes residing in neighboring genomic regions that represent the other driving forces. Therefore, the candidate genes identified that were found to be associated with adaptive radiation do not necessarily point towards causal evolutionary mechanisms, they may also be the by-products of the long-term process of adaptive radiation. In addition, given that CG methylation is the main cytosine methylation type in plant species, we limited our analysis to CG methylation because the uncertainty remains regarding the number of candidate genes and functional roles of CHG and CHH methylation. We anticipate that future studies with larger sample sizes will be able to improve the statistical power and investigate *trans*-genetic control.

## 5. Conclusions

Our study confirms that the ten *Aquilegia* species from Asia, Europe and North America formed three phylogenetically independent lineages corresponding to their geographic origins. We further demonstrated that the candidate genes involved in adaptive speciation are functionally enriched in the pathways related to cell reproduction (e.g., telomere maintenance), stress tolerance (e.g., response to wounding) and basic cellular activities, which indicate that specific genetic determinants related to chromosomal architecture might have played a role in the speciation of the *Aquilegia* species. At the same time, functional enrichment analyses identified significant associations with adaptation-related traits and these features exhibited that epigenetic factors may also play a role in the responses to diverse environmental conditions. In addition, we conclude that many CG-loss variations can directly lead to the depletion of CG methylation and many DMRs are not affected by any *cis*-variations. Since gene body CG methylation in plants generally stabilizes gene expression and is positively correlated with gene expression, differential methylation in our study ostensibly indicates differing amounts of gene products. These results are consistent with previous studies and provide an exploratory overview of how genetic and epigenetic signatures are associated with the diversification of the *Aquilegia* species. 

## Figures and Tables

**Figure 1 genes-13-00793-f001:**
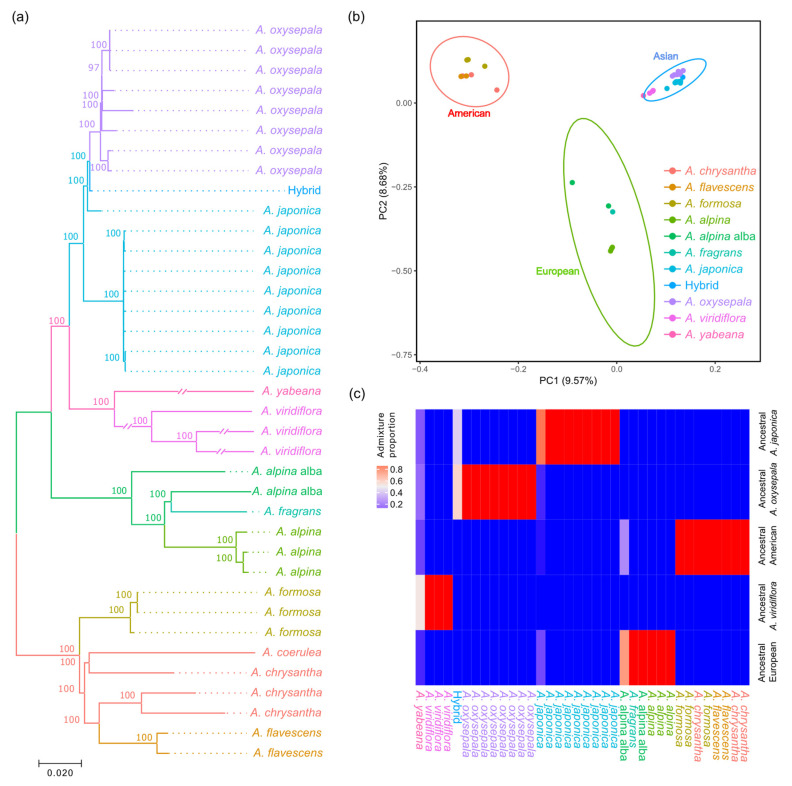
Phylogenetic relationship and population structure of the ten worldwide *Aquilegia* species. (**a**) Phylogenetic tree of the 36 accessions, constructed by a neighbor-joining algorithm based on 689,123 whole-genome SNPs. (**b**) PCA reveals genetic similarity within each of the three lineages and genetic disparity between the lineages based on 15,988 LD-pruned SNPs. Ellipses of each lineage denote a 99% confidence region estimated from the distribution of the first two principal components. (**c**) Population admixture of the 36 *Aquilegia* accessions.

**Figure 2 genes-13-00793-f002:**
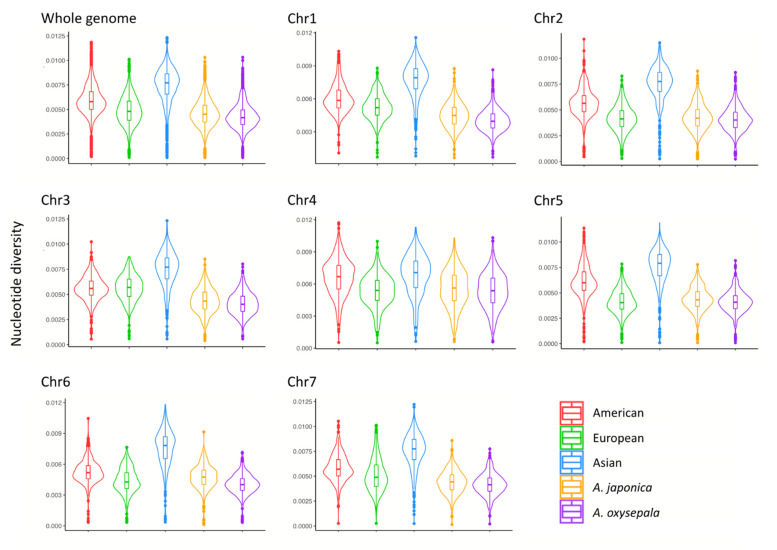
Distribution of nucleotide diversity (π) at the whole-genome level and the per-chromosome level. Nucleotide diversity was estimated for each lineage pooling corresponding species, as well as for *A. japonica* and *A. oxysepala*.

**Figure 3 genes-13-00793-f003:**
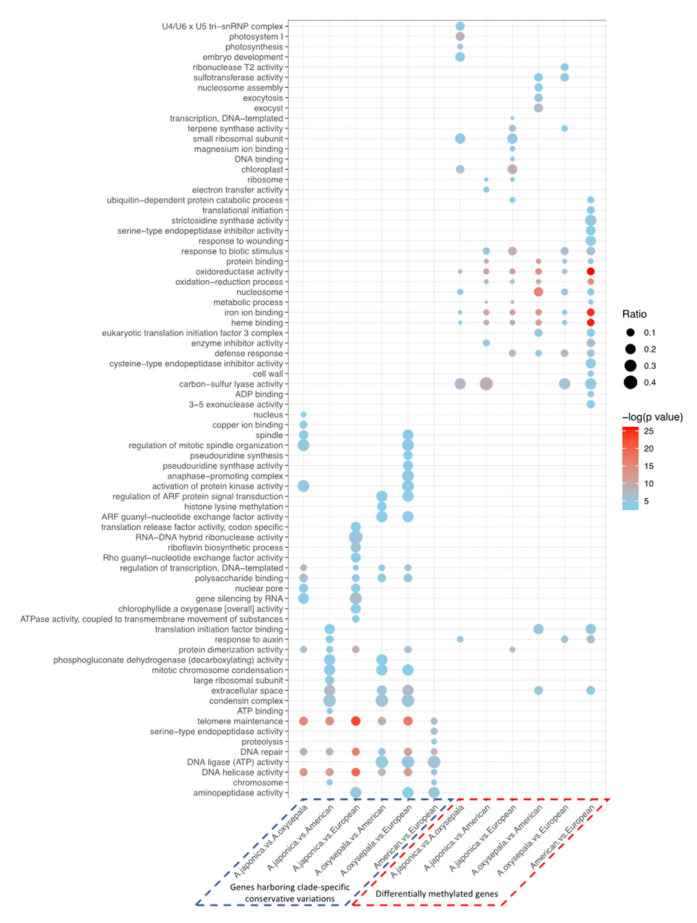
Functional enrichment of the genes harboring highly impactful CCVs and DMGs. CCV-containing genes, specific to either of the two lineages/species being compared, were merged to construct a target gene set. Ratio denotes proportion of CCV-containing genes or DMGs in the corresponding gene set of interest. Absence of dot indicates no significant enrichment.

**Figure 4 genes-13-00793-f004:**
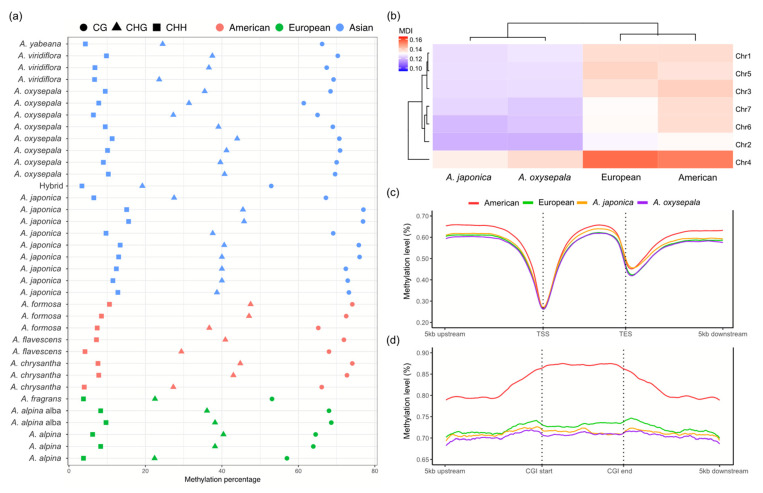
Patterns of cytosine methylation for the ten worldwide *Aquilegia* species. (**a**) Genome-wide cytosine methylation levels of 36 accessions. (**b**) MDI illustrates chromosome-level CG methylation similarity. *Aquilegia viridiflora* was used as the reference. (**c**) CG methylation profiling in genic region across the four *Aquilegia* groups. Each row represents one genic region starting at 5-kb upstream of its TSS and terminating at 5-kb downstream of its TES, sorted by mean methylation level of all the analyzed CG loci. Gene body regions were scaled to have the same length. (**d**) CG methylation profiling in and around the CG islands.

**Figure 5 genes-13-00793-f005:**
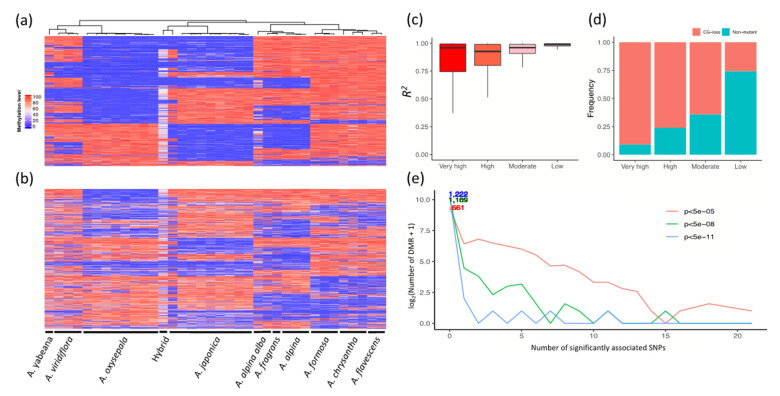
Association between the CG-loss variations and epigenetic variability. (**a**) Top 3000 most variable CG loci containing CG-loss variations. (**b**) Top 3000 most variable non-variant CG loci across 36 accessions that show clade-specific methylation patterns. CG methylation in the hybrids tends to be neutralized, possibly due to heterozygosity. (**c**) Linear regression demonstrates that CG-loss variations explain a large proportion of CG methylation variation. (**d**) Summary of composition of each category regarding whether each CG locus contains a CG-loss variation. Epigenetic variability was determined by standard deviation in methylation β value across all 36 accessions. CG loci with the top 10,000, 10,001–50,000 and 50,001–150,000 largest standard deviation was ordinally labelled as possessing “very high”, “high” and “moderate” variability, respectively. The rest of the CG loci were labelled as possessing “low” variability. (**e**) Association test shows most DMRs were independent of cis-acting SNPs. Results under different significance levels are compared in this exploratory analysis.

**Table 1 genes-13-00793-t001:** Information of the high-impact conservative clade-specific variants (CCVs) in the cell reproduction related genes.

Gene	Variant-Carrying	Reference	Chromosome	Position	Reference Allele	Variant	Annotation	Gene Function
*Aqcoe1G273400*	Asian	American	Chr1	18994915	GAA	GAAA	frameshift	DNA mismatch repair protein *MutS2*
*Aqcoe2G151500*	European	American	Chr2	15305837	A	G	splicing	PIF1-like helicase
	European	American		15307442	A	C	stop gain	
	European	American		15309865	AATATATAT	AATATATATAT	frameshift	
	European	Asian		15307442	A	C	stop gain	
	European	Asian		15309865	AATATATAT	AATATATATAT	frameshift	
	*A. oxysepala*	*A. japonica*		15305837	A	G	splicing	
	*A. oxysepala*	*A. japonica*		15309267	AT	A	frameshift	
*Aqcoe2G177700*	European	American	Chr2	21794397	TATGCACCAAAGGTATCACGATGC	TATGC	frameshift	PIF1-like helicase
	European	American		21794979	TT	TTGT	frameshift	
	European	Asian		21794397	TATGCACCAAAGGTATCACGATGC	TATGC	frameshift	
	*A. oxysepala*	*A. japonica*		21795089	CA	C	frameshift	
*Aqcoe6G208600*	European	American	Chr6	15364081	A	ATCTCTTCG	frameshift	PIF1-like helicase
	European	Asian		15364081	A	ATCTCTTCG	frameshift	
	*A. japonica*	*A. oxysepala*		15364330	TAA	TA	frameshift	
*Aqcoe6G253800*	European	American	Chr6	22789898	C	T	stop gain	DNA helicase
	European	American		22790012	G	A	splicing	
	European	Asian		22789898	C	T	stop gain	
	*A. japonica*	*A. oxysepala*		22790012	G	A	splicing	
*Aqcoe2G276600*	Asian	American	Chr2	33314422	AGGGGG	AGGGGGG	frameshift	DNA mismatch repair protein *Msh6*
*Aqcoe6G160300*	*A. japonica*	*A. oxysepala*	Chr6	9414625	G	A	stop gain	*TPX2*
*Aqcoe7G062500*	*A. oxysepala*	*A. japonica*	Chr7	3789055	G	A	stop gain	cell cycle-regulated microtubule-associated protein

**Table 2 genes-13-00793-t002:** Significant correlation between differential methylation and natural selection.

Type of Selection	Differential Methylation	Jap-Oxy *	Jap-Ame	Jap-Eur	Oxy-Ame	Oxy-Eur	Ame-Eur
	DMG	7.2%	7.3%	11.9%	6.7%	8.4%	8.9%
Positive selection	non-DMG	4.4%	5.0%	5.6%	4.7%	5.4%	5.4%
	*p* value	0.11	7.3 × 10^−2^	3.9 × 10^−5^	6.7 × 10^−2^	1.8 × 10^−2^	2.8 × 10^−4^
	DMG	3.1%	1.8%	2.4%	2.3%	2.0%	1.9%
Purifying selection	non-DMG	4.3%	4.3%	4.7%	4.9%	5.1%	4.0%
	*p* value	0.53	3.2 × 10^−2^	9.1 × 10^−2^	1.3 × 10^−2^	8.4 × 10^−3^	1.0 × 10^−2^

*: Each percentage represents the proportion of genes belonging to either DMGs or non-DMGs compared between the two corresponding clades that are under corresponding or higher strength of positive selection. For example, 7.2% indicates that 7.2% DMGs compared with *A. japonica* and *A. oxysepala* are under strong selection; 4.4% indicates that 4.4% non-DMGs compared with these two species are under strong selection. *p* values were obtained from Chi-square tests and were not adjusted for multiple testing, due to the dependence arising from overlapping gene sets. Jap: *A. japonica*; Oxy: *A. oxysepala*; Ame: American; Eur: European.

## Data Availability

The data presented in this study are available in the Appendix A. All data generated from the study were submitted to EBI, under the accession number PRJEB34182.

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
