# Peer review of "Genetic and Epigenetic Signatures Associated with the Divergence of Aquilegia Species"

_genes, 2022, doi:10.3390/genes13050793_

Round 1

Reviewer 1 Report

Dear Authors,

The manuscript titled “Genetic and epigenetic signatures associated with the divergence of Aquilegia species” address an important issue of study the phylogeny and population structure of the model system in adaptive radiation research which is genus Aquilegia (columbine). The research was carefully planned, and the authors chose the appropriate statistical analyzes. Obtained results allowed to  identified candidate genes exhibiting lineage-specific genetic or epigenetic variation patterns that were signatures of inter-specific divergence. Authors suggested that epigenetic modification may be a complementary mechanism facilitating phenotypic diversity of the Aquilegia species. Moreover, the authors were aware of the limitations of their own results resulting from a small research sample and not a fully comprehensive analysis of changes in the methylome of the studied species. Authors captured these drawbacks in the section "Limitations and future directions". Which proves the authors' scientific expertise. The only thing I miss at work is a more comprehensive discussion of the results obtained with the literature data in the field of epigenetic modification as a complementary mechanism facilitating phenotypic diversity. What it look situation I other species? In my opinion improving discussion will allow to increasing the scientific quality of this manuscript. Beneficial will be also adding chapter conclusions. 

Reviewer 2 Report

In this manuscript (genes-1678125) entitled "Genetic and epigenetic signatures associated with the divergence of Aquilegia species" submitted to Genes, authors charaterized the genetic and epigenetic signatures are associated with the diversification of Aquilegia species. Through surveying the genomes and DNA CG methylomes of ten Aquilegia species covering the Asian, European and North American lineages, authors found high genetic and DNA methylomic divergence across these three lineages, and identified candidate genes exhibiting lineage-specific genetic or epigenetic variation patterns that were signatures of inter-specific divergence. Authors demonstrated that these species-specific genetic variations and epigenetic variabilities are partially independent and are both functionally related to various biological processes vital to adaptation, including stress tolerance, cell reproduction and DNA repair.

This study is an interesting work, I have however several concerns that may be addressed to improve the quality of the work.

  1. DNA methylation could occur at the sequence context of CG, CHG and CHH. It is disappointing that authors did not survey the CHG and CHH methylation, which should be performed in the revision as a whole story.
  2. DNA methylation and their function in genome evolution should be introduced in the introduction section, please revise.
  3. DNA methylomes vary among plant species, developmental stages and environmental conditions, which should discussed in the revision.
  4. Full names of the abbreviations PCA, DMG, CCV, TPX2, MSH, and ALX1 should be spelt out at their first appearance in this article.

Round 2

Reviewer 2 Report

Authos have addressed my concerns in the revised manuscript.